# Evaluation of New Potential Inflammatory Markers in Patients with Nonvalvular Atrial Fibrillation

**DOI:** 10.3390/ijms24043326

**Published:** 2023-02-07

**Authors:** Gabriela Lopes Martins, Rita Carolina Figueiredo Duarte, Érica Leandro Marciano Vieira, Natália Pessoa Rocha, Estêvão Lanna Figueiredo, Francisco Rezende Silveira, José Raymundo Sollero Caiaffa, Rodrigo Pinheiro Lanna, Maria das Graças Carvalho, András Palotás, Cláudia Natália Ferreira, Helton José Reis

**Affiliations:** 1Departamento de Farmacologia, ICB, Universidade Federal de Minas Gerais, Belo Horizonte 31270-901, MG, Brazil; 2Neuropsychiatry Program, Department of Psychiatry and Behavioral Sciences, McGovern Medical School, University of Texas Health Science Center, Houston, TX 77054, USA; 3Hospital Lifecenter, Belo Horizonte 30110-921, MG, Brazil; 4Hospital Semper, Belo Horizonte 30130-110, MG, Brazil; 5Centro de Especialidades Médicas Ipsemg, Belo Horizonte 30150-240, MG, Brazil; 6Asklepios-Med, 6722 Szeged, Hungary; 7Institute of Fundamental Medicine and Biology, Kazan Federal University, 420008 Kazan, Russia

**Keywords:** atrial fibrillation, inflammation, biomarkers, inflammatory mediators, arrhythmia

## Abstract

Atrial fibrillation (AF), the most common arrhythmia in clinical practice, is associated with an increase in mortality and morbidity due to its high potential to cause stroke and systemic thromboembolism. Inflammatory mechanisms may play a role in the pathogenesis of AF and its maintenance. We aimed to evaluate a range of inflammatory markers as potentially involved in the pathophysiology of individuals with nonvalvular AF (NVAF). A total of 105 subjects were enrolled and divided into two groups: patients with NVAF (n = 55, mean age 72 ± 8 years) and a control group of individuals in sinus rhythm (n = 50, mean age 71 ± 8 years). Inflammatory-related mediators were quantified in plasma samples by using Cytometric Bead Array and Multiplex immunoassay. Subjects with NVAF presented significantly elevated values of interleukin (IL)-2, IL-4, IL-6, IL-10, tumor necrosis factor (TNF), interferon-gamma, growth differentiation factor-15, myeloperoxidase, as well as IL-4, interferon-gamma-induced protein (IP-10), monokine induced by interferon-gamma, neutrophil gelatinase-associated lipocalin, and serum amyloid A in comparison with controls. However, after multivariate regression analysis adjusting for confounding factors, only IL-6, IL-10, TNF, and IP-10 remained significantly associated with AF. We provided a basis for the study of inflammatory markers whose association with AF has not been addressed before, such as IP-10, in addition to supporting evidence about molecules that had previously been associated with the disease. We expect to contribute to the discovery of markers that can be implemented in clinical practice hereafter.

## 1. Introduction

Atrial fibrillation (AF) is the most common sustained cardiac arrhythmia in the adult population and has an estimated prevalence between 2% and 4%, with its incidence rising with aging [1,2]. AF is associated with major morbidity and mortality, as it increases the risk for systemic embolism and stroke, leading to a significant burden on patients, society, and the health economy [3,4].

Inflammatory processes have been implicated in the pathogenesis of AF, both as a cause and consequence of the disease [5]. Patients with AF generally have comorbidities such as hypertension, diabetes, heart failure, coronary artery disease, chronic kidney disease, obesity, and obstructive sleep apnea [6,7,8,9]. These, together with modifiable risk factors (smoking, excessive alcohol consumption, sedentary lifestyles, and extreme exercise) are associated with increased levels of circulating cytokines [2,10]. In this scenario, high levels of plasma inflammatory mediators can cause the electrical and structural cardiac remodeling observed in AF. However, once arrhythmia is established, inflammatory processes also occur in response to endothelial lesions in the atrial tissue, sustaining the disease by the same remodeling mechanism, which generated the expression “AF begets AF” [5].

Different inflammatory mediators have been regarded as both diagnostic and prognostic markers of AF, including C-reactive protein (CRP), which was the first inflammatory marker whose association with AF was demonstrated in a study that an increase in the development of AF after coronary artery bypass grafting (CABG), coinciding with peak concentrations of serum CRP, was observed [11]. Next, higher levels of CRP were detected in peripheral blood samples from individuals with AF who did not undergo surgery, compared with controls [12]. CRP was also associated with AF recurrence after pharmacological cardioversion [13] and, more recently, after catheter ablation therapy [14].

Interleukin (IL)-6 is another inflammatory mediator widely assessed in AF, and higher levels of this cytokine were observed in the serum of patients with AF [15], as well as in samples obtained from the left atrium, coronary sinus, and femoral artery and vein [16] compared with individuals in sinus rhythm. Moreover, higher levels of IL-6 were associated with the development of AF after CABG [17], the recurrence of AF after ablation [18], and a greater chance of stroke in patients with arrhythmia [19,20]. More recently, a study showed that in conditions where systemic inflammation is installed, presenting with high levels of IL-6, atrial remodeling can quickly be induced through the downregulation of cardiac connexins, increasing the risk of developing AF and other related complications [21].

Concerning other cytokines, IL-2 was associated with the development of AF in patients with cardiac comorbidities [22] and after CABG surgery [23], as well as considered a predictive factor for arrhythmia recurrence after catheter ablation therapy [24]. Higher levels of IL-10 and tumor necrosis factor (TNF) were found in patients with AF, compared with controls [15], and the latter was also suggested as a predictive marker of stroke in patients with NVAF [25]. Furthermore, TNF demonstrated an important role in electrical and structural atrial remodeling, and the use of anti-TNF agents has been proposed as a potential therapeutic target for AF [26]. IL-10 and interferon-gamma (IFN-γ) were associated with the development of arrhythmia after CABG surgery [23], and in a cohort study, the second was considered a marker for stroke prediction and all-cause mortality in patients with new-onset AF [27]. Additionally, an interventional study with patients with permanent cardiac pacemaker implantation showed that the treatment with metoprolol for six months resulted in a decrease in the blood levels of highly sensitive CRP, IL-6, and TNF. These findings were considered of great clinical importance since they were associated with a lower incidence of AF and an improvement in the quality of life in these patients [28].

Among these inflammatory mediators, it is also worth mentioning growth differentiation factor (GDF)-15, which was related to the incidence of AF [29] and considered a risk factor for bleeding, mortality, and stroke in patients with AF undergoing oral anticoagulants, compared with those without these conditions [30,31]. Moreover, in patients with NVAF that were nonanticoagulated, elevated serum levels of GDF-15 were considered a risk for developing left atrial thrombus [32].

Despite all this evidence, however, none of these markers are currently used in clinical practice in the management of AF. Therefore, the current study aimed to evaluate a range of inflammatory markers in patients with nonvalvular AF (NVAF), which may serve as new targets for NVAF studies.

## 2. Results

### 2.1. Demographic, Clinical, and Laboratory Characteristics

The demographic, clinical, and laboratory characteristics of participants are shown in Table 1. There were no differences between groups regarding age, sex, and statin use. The frequency of hypertension and type II diabetes mellitus was higher among patients with AF in comparison with controls. In addition, higher levels of ALT, AST, GGT, creatinine, and uric acid were found in patients with AF, compared with controls.

### 2.2. Comparison of Inflammatory Mediators between the Groups

Patients with NVAF presented elevated plasma levels of IL-2, IL-4, IL-6, IL-10, TNF, IFN-γ, IP-10, and MIG (Figure 1, Appendix A) in comparison with controls. Higher plasma levels of GDF-15, MPO, NGAL, and SAA were also observed in individuals with AF, in relation to controls (Figure 2, Appendix A).

No significant differences were found when comparing the groups for the parameters RANTES, MCP-1, IL-8, TGF-β, ADAMTS13, myoglobin, sICAM-1, p-selectin, and sVCAM-1 (Appendix A).

### 2.3. Logistic Regression Models

In the univariate analysis of inflammatory mediators that showed higher levels in patients with AF, compared to controls, significant associations were observed for IL-2, IL-4, IL-6, IL-10, TNF, IFN-γ, and IP-10 (Appendix A). These markers remained significantly associated with AF after adjusting for age, sex, hypertension, and diabetes mellitus (Model 1), with the exception of IL-2. On the other hand, when adjusting for age, sex, hypertension, diabetes mellitus, ALT, AST, GGT, creatinine, and uric acid (Model 2), only IL-6, IL-10, TNF, and IP-10 levels remained significantly associated with AF. The odds ratio (OR) and the confidence intervals calculated in the multivariate regression analyses are shown in Table 2 (for model validation, see Appendix A).

## 3. Discussion

Herein, we evaluated plasma levels of different inflammatory mediators in patients with NVAF. We observed that NVAF patients have significantly higher plasma levels of markers whose association with AF was poorly or even unreported in the literature, such as IL-4, IP-10, MIG, NGAL, and SAA. In addition, we also observed elevated levels of molecules that had been previously associated with AF, such as IL-6, IL-2, IL-10, TNF, IFN-γ, and GDF-15. When analyzing these parameters, it is necessary to consider the presence of other clinical conditions in which the levels of inflammatory mediators may also be increased, such as hypertension and diabetes. In fact, we found that patients with NVAF had a higher frequency of both conditions when compared with controls. These results are corroborated by what is observed in clinical practice since hypertension and diabetes are the main comorbidities in patients with AF [33,34]. Therefore, we performed a multivariate regression analysis in order to verify if the inflammatory parameters that were higher in the group of patients are associated with AF, adjusting for age, sex, hypertension, and diabetes mellitus (Model 1). In this analysis, significant associations were observed for IL-2, IL-4, IL-6, IL-10, TNF, IFN-γ, and IP-10.

Nevertheless, it is worth mentioning that our patients had higher levels of liver transaminases (ALT, AST, and GGT), creatinine, and uric acid, in comparison with controls, factors that can also contribute to the increase in circulating inflammatory mediators. Several studies have demonstrated that elevated levels of liver enzymes are associated with AF. In a study that evaluated participants in the Framingham Heart Study Original and Offspring cohorts, AST was associated with a 10-year risk of developing AF [35]. In another prospective study, high serum levels of GGT in individuals who developed AF were associated with the incidence of arrhythmia [36]. In patients with coronary heart disease, the onset of AF was also associated with high levels of GGT, proposing the use of this enzyme as a risk marker for arrhythmia [37]. Furthermore, in patients with paroxysmal AF undergoing catheter ablation therapy, high levels of GGT were found in individuals who had arrhythmia recurrence, compared to those who maintained sinus rhythm after therapy. Based on this, GGT has been associated with the recurrence of AF, and its quantification was suggested to assess patients who are at higher risk of recurrence after ablation therapy, in those in whom this approach is indicated [38].

Regarding renal function, we found that patients with NVAF had significantly higher creatinine levels compared with controls. However, patients with AF were within the typical range for serum creatinine [39]. Considering that AF is a disease that mainly affects the elderly and that renal function gradually reduces with age, it is expected that these two clinical conditions can coexist. Furthermore, like AF, renal dysfunction is also associated with the risk of developing thromboembolic events. Thus, monitoring renal function in patients with AF deserves special attention [40]. Noteworthy, although we found elevated levels of hepatic and renal markers in patients with NVAF, they did not fulfill the diagnostic criteria for liver or kidney disease.

Regarding the higher levels of uric acid observed in patients with NVAF in comparison with controls, our data corroborate a previous study that has reported a high incidence rate of arrhythmia in patients with gout [41]. Moreover, in another cohort, the prevalence of AF was higher in individuals who had hyperuricemia in comparison with those who did not develop this condition. Thus, it has been suggested that therapies to reduce uricemia may play a role in the prevention and treatment of AF [42].

As mentioned earlier, all these factors (i.e., elevated levels of hepatic transaminases, creatinine, and uric acid) are associated with a chronic inflammatory state and, consequently, with increased levels of circulating inflammatory mediators. Therefore, we performed a multivariate logistic regression analysis for each inflammatory mediator that was previously significant, adjusting also for these factors (Model 2), in addition to the other confounders (age, sex, hypertension, and diabetes mellitus). In this analysis, only IL-6, IL-10, TNF, and IP-10 remained significantly associated with AF.

As mentioned above, IL-6, IL-10, and TNF have been widely associated with AF in previous studies. However, in addition to corroborating the already existing data, we would like to emphasize that in our study IP-10 was associated with AF after adjusting for clinical confounders. IP-10 (also known as C-X-C motif chemokine ligand 10, CXCL10) is a chemokine with pro-inflammatory and anti-angiogenic properties, of which the role in the connection between inflammation and angiogenesis has been previously proposed [43]. The involvement of IP-10 with atherosclerosis and coronary syndromes has also been suggested [44], and in recently published cohort data, IP-10 has been shown to have a role in heart failure and its associated mortality in African Americans [45]. Over the last two decades, different studies, both experimental and clinical, have already tried to understand the role of IP-10 in cardiovascular diseases, but its actions are complex and seem to be associated with the type of receptor it binds to [46]. We recently reported higher levels of IP-10 in patients with AF using rivaroxaban, in comparison with controls [47]. However, to our knowledge, there is no other evidence associating this chemokine directly with AF. Given the above, perhaps we can be facing a promising marker for AF that deserves further investigation in future studies.

It is worth mentioning that the findings of our study should be interpreted considering its limitations. First, the results cannot be extrapolated to patients with AF conditions different from those included in this study, for example, patients with valvular AF. In addition, due to the relatively small sample size and the large number of parameters that were evaluated, there may be an implication in the accuracy of the results found in the regression analyses. Lastly, this is a cross-sectional study, and longitudinal studies are needed for assessing predictive factors. Thus, we emphasize the achievement of prospective studies with a larger sample size to validate the results herein observed.

## 4. Materials and Methods

### 4.1. Study Population

A total of 105 participants were enrolled (55 patients with NVAF and 50 controls). Eligible participants comprised individuals with a history of AF documented by electrocardiogram within the 12 months before collection, and for whom chronic oral anticoagulation was indicated (CHA_2_DS_2_-VASC ≥ 2). The patients were enrolled from the outpatient clinics of the hospitals of Lifecenter, Semper, and Ipsemg (Belo Horizonte, Minas Gerais, Brazil) during the period from October 2013 to January 2017. A group of controls composed of individuals in sinus rhythm with no previous diagnosis of AF or use of any anticoagulant therapy was recruited from the local community.

Participants were excluded if they had used any antiplatelet agent, non-steroidal anti-inflammatory drugs, heparin, hormone replacement therapy, antifibrinolytics, amiodarone, verapamil, quinidine, azole antifungals, or ritonavir in the 4 weeks prior to the study. Subjects were also excluded if they presented a current diagnosis of an alcohol use disorder, chronic kidney disease (creatinine clearance < 30 mL/min), severe dyslipidemia, bleeding disorders, liver and thyroid diseases, infectious, inflammatory, autoimmune, or malignant diseases, pregnancy, puerperium, and breastfeeding.

### 4.2. Biological Samples

Peripheral blood samples were drawn by venipuncture into tubes containing ethylenediaminetetraacetic acid and into tubes without anticoagulants. The samples were processed within 4 h of collection by centrifugation at 1100× *g*, 25 °C, for 15 min, aliquoted, and stored at −80 °C until analysis.

### 4.3. Laboratory Characterization

CRP, total cholesterol, high-density lipoprotein (HDL), low-density lipoprotein (LDL), triglycerides, aspartate transaminase (AST), alanine aminotransferase (ALT), gamma-glutamyl transferase (GGT), creatinine, and uric acid were assessed in serum samples, by using Vitros 250 system, Johnson & Johnson^®^ in the Laboratory of Hospital Risoleta Tolentino Neves (Belo Horizonte, MG, Brazil). CRP was reported as a categorical variable once ultra-sensitive analysis was not performed. White blood cell (WBC) count was assessed in whole blood samples by using the Coulter T-890^®^ hematological analyzer.

### 4.4. Analysis of Inflammatory Mediators

The inflammatory markers were assessed in plasma samples by Cytometric Bead Array (CBA, BD Bioscience, San Diego, CA, USA), using the following kits: Cytokines Th1/Th2 (IL-2, IL-4, IL-6, IL-10, TNF and interferon-gamma (IFN-γ)); Chemokines (CXCL8/IL-8, CCL5/regulated on activation, normal T cell expressed and secreted (RANTES), CXCL9/monokine induced by interferon-gamma (MIG), CCL2/monocyte chemoattractant protein (MCP)-1, CXCL10/interferon-gamma-induced protein (IP)-10); and human transforming growth factor-beta (TGF-β)-1 Single Plex Flex Set. A multiplex immunoassay (Merck Millipore, Billerica, MA, USA) was used to assess the levels of proteins of the Human Cardiovascular Disease-Panel II kit (disintegrin and metalloproteinase with thrombospondin type 1 motif, 13 (ADAMTS13), growth differentiation factor (GDF)-15, myoglobin, soluble intercellular adhesion molecule (sICAM)-1, myeloperoxidase (MPO), p-selectin, neutrophil gelatinase-associated lipocalin (NGAL), soluble vascular cell adhesion protein (sVCAM)-1, serum amyloid A (SAA)).

### 4.5. Statistical Analysis

Differences between categorical variables were evaluated by the Pearson Chi-square test or Fisher’s exact test, when appropriate. Continuous variables were first tested by the Shapiro–Wilk normality test. Two groups (NVAF vs. controls) were compared using the Student’s *t*-test or the Mann–Whitney *U* test, when data were determined whether to follow or not a normal distribution, respectively. Logistic regression models were performed for each inflammatory mediator that had significantly higher levels in patients with NVAF when compared to controls. Firstly, a univariate model was used to verify the association (odds ratio) of each marker with AF, without any adjustment. The variables that showed a significant result in the univariate analysis were then included in a multivariate analysis, for which two models were used to evaluate the association of each inflammatory mediator with AF. In the first (Model 1), the variables were adjusted for age, sex, hypertension, and diabetes mellitus, and in the second (Model 2), the adjustment was made to age, sex, hypertension, diabetes mellitus, ALT, AST, GGT, creatinine and uric acid. All statistical tests were two-tailed and a significance level of *p* = 0.05 was set.

## 5. Conclusions

Inflammatory mechanisms have been associated with the pathophysiology of AF for more than two decades. Despite great efforts, several inflammatory markers were suggested as predictive markers for arrhythmia, but none is yet used effectively in clinical practice. Our results provide a basis for the study of inflammatory markers that had not yet been well addressed in AF, especially IP-10, besides supporting evidence about molecules that had previously been associated with the disease. Thus, although there is still a need to conduct prospective studies with a larger number of participants, we are hopeful these findings may help in the identification of novel inflammatory markers potentially involved in the pathophysiology of NVAF.

## Figures and Tables

**Figure 1 ijms-24-03326-f001:**
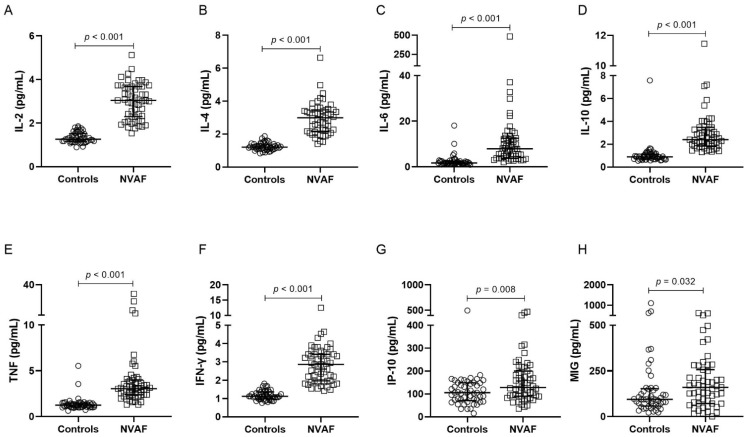
Plasma levels of inflammatory mediators IL-2 (**A**), IL-4 (**B**), IL-6 (**C**), IL-10 (**D**), TNF (**E**), IFN-γ (**F**), IP-10 (**G**), and MIG (**H**) comparing controls (n = 50) and NVAF (n = 55) groups. The dosage of these markers was performed by Cytometric Bead Array. The groups were compared by the Mann–Whitney *U* test. The horizontal bars show the median and the interquartile range. NVAF = nonvalvular atrial fibrillation; IL = interleukin; TNF = tumor necrosis factor; IFN-γ = interferon-gamma; IP-10 = interferon-gamma-induced protein; MIG = monokine induced by interferon-gamma.

**Figure 2 ijms-24-03326-f002:**
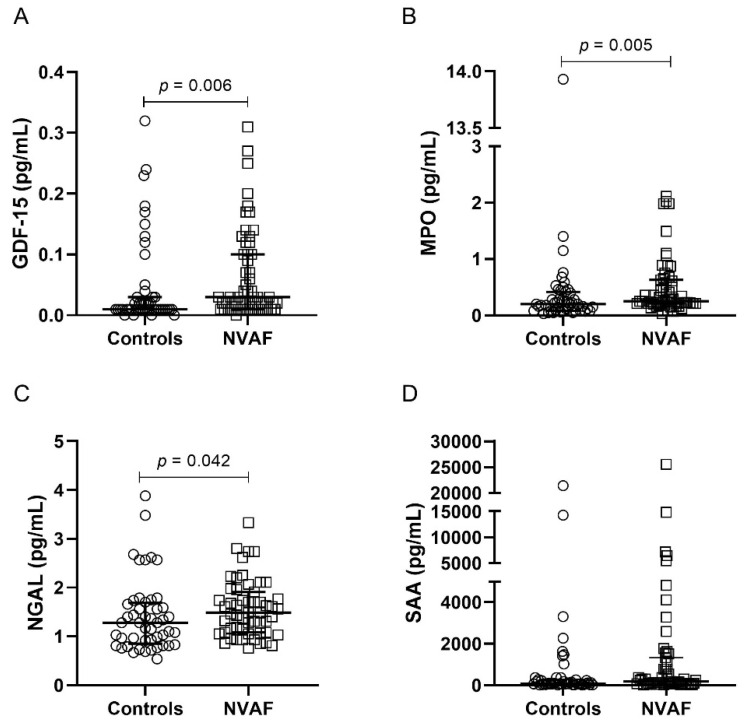
Plasma levels of inflammatory mediators GDF-15 (**A**), MPO (**B**), NGAL (**C**), and SAA (**D**), comparing controls (n = 50) and NVAF (n = 55) groups. The dosage of these markers was performed by Multiplex immunoassay. The groups were compared by the Mann–Whitney *U* test. The horizontal bars show the median and the interquartile range. NVAF = nonvalvular atrial fibrillation; GDF = growth differentiation factor; MPO = myeloperoxidase; NGAL = neutrophil gelatinase-associated lipocalin; SAA = serum amyloid protein.

**Table 1 ijms-24-03326-t001:** Demographic, clinical, and laboratory characterization of participants of the study.

Parameters	Controls (N = 50)	NVAF (N = 55)	*p*-Value
Age, mean ± SD	71 ± 8	72 ± 8	0.822
Sex			
Female, N (%)	26 (52%)	28 (50.9%)	0.911
Male, N (%)	24 (48%)	27 (49.1%)	
Hypertension, N (%)	29 (58%)	53 (96.4%)	**<0.001**
Diabetes mellitus 2, N (%)	5 (10%)	20 (36.4%)	**0.002**
Statins, N (%)	21 (42%)	31 (58.5%)	0.094
CHA_2_DS_2_-VASc, median (IQR)	-	4 (1)	-
Total cholesterol (mg/dL), mean ± SD	181 ± 36	173 ± 39	0.227
LDL (mg/dL), mean ± SD	98 ± 29	87 ± 34	0.080
HDL (mg/dL), mean ± SD	54 ± 14	55 ± 15	0.682
Triglycerides (mg/dL), median (IQR)	130.5 (72.5)	122.5 (117.5)	0.992
ALT (U/L), median (IQR)	24 (10)	26.5 (15.5)	**0.030**
AST (U/L), median (IQR)	26 (7.25)	30 (12.75)	**0.023**
GGT (U/L), median (IQR)	23.5 (11.5)	41.5 (33.5)	**<0.001**
Creatinine (mg/dL), median (IQR)	1 (0.3)	1.1 (0.3)	**0.031**
Uric acid (mg/dL), median (IQR)	5.45 (1.75)	6.2 (2.65)	**0.002**
WBC, median (IQR)	5200 (2520)	5750 (2040)	0.438
CRP ≥ 10 mg/L, N (%)	8 (16.3%)	12 (21.8%)	0.478

Student’s *t*-tests were performed for normal continuous variables (values expressed as mean and standard deviation); Mann–Whitney *U* test for non-normal continuous variables (values expressed as median and interquartile range); and Pearson Chi-Square test for categorical variables (values expressed as absolute and relative frequency). ALT = Alanine aminotransferase; AST = Aspartate transaminase; CHA_2_DS_2_-VASc = Congestive heart failure, Hypertension, Age ≥ 75 (doubled), Diabetes, Stroke (doubled), Vascular disease, Age 65–74, e Sex category (female); CRP = C-reactive protein; GGT = Gamma-glutamyltransferase; HDL = High-density lipoprotein; IQR = Interquartile range; LDL = Low-density lipoprotein; NVAF = Nonvalvular atrial fibrillation; SD = Standard deviation; WBC = White blood cells. Statistically significant differences between groups are indicated in bold, with a significance level of *p* < 0.05.

**Table 2 ijms-24-03326-t002:** Multivariate logistic regression for inflammatory mediators.

	Model 1		Model 2	
Inflammatory Mediators (pg/mL)	OR (95% CI)	*p*-Value	OR (95% CI)	*p*-Value
IL-2	2.54 × 10^12^ (0.49–1.32 × 10^25^)	0.056	9.58 × 10^88^ (0.00–)	0.981
IL-4	2.80 × 10^4^ (38.90–2.01 × 10^7^)	**0.002**	3.61 × 10^174^ (0.00–)	0.959
IL-6	1.61 (1.26–2.05)	**<0.001**	2.85 (1.56–5.23)	**0.001**
IL-10	8.99 (3.33–24.31)	**<0.001**	6.17 (2.33–16.33)	**<0.001**
TNF	8.33 (3.32–20.87)	**<0.001**	12.24 (3.42–43.76)	**<0.001**
IFN-γ	4.91 × 10^10^ (2.01–1.20 × 10^21^)	**0.044**	3.42 × 10^42^ (0.00–)	0.984
IP-10	1.01 (1.00–1.02)	**0.007**	1.01 (1.00–1.02)	**0.046**

Model 1: variables adjusted for age, sex, hypertension, and diabetes mellitus. Model 2: variables adjusted for age, sex, hypertension, diabetes mellitus, ALT, AST, GGT, creatinine, and uric acid. CI = confidence interval; IFN-γ = interferon-gamma; IL = interleukin; IP-10 = interferon-gamma-induced protein; OR = odds ratio; TNF = tumor necrosis factor. Statistically significant differences are indicated in bold, with a significance level of *p* < 0.05.

## Data Availability

Details about the materials and protocols used in this research are available upon request to the corresponding author. The data sets used in this study are not available to the public, due to the confidentiality of the participants’ data guaranteed with the approval of the project by the ethics committee. If desired, requests for access to the data sets should be sent to the corresponding author.

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
