# Peer review of "Evaluation of New Potential Inflammatory Markers in Patients with Nonvalvular Atrial Fibrillation"

_ijms, 2023, doi:10.3390/ijms24043326_

Round 1
Reviewer 1 Report
Thank you very much for the opportunity of reviewing this paper. AF is one of the most frequente cardiac arrhythmia and so all that helps clinician to understand AF is absolutely welcome. I read this paper with interest but I have some concerns.
1) the sample size is rather small if you consider that AF is very common. You should enlarge your population to be more appealing
2) It's not clear form the methods in what time frame you enrolled the patients. This information is important to get the reader to understand is the sample makes sense.
3) In the methods (line 103) you stated that patients were selected. This is a rather dangerous statement in a scientific paper.
4) It's not clear in the methods why you run two different models of multivariate analysis. I guess because the sample is too small and it prevents you to correct for too many variables all at once. Please explain better.
5) You should check for correlations among the different ILs. It is possibile they are correlated. If it is so it would be not surprising that different ILs perform similarly in the models.
6) This paper, besides having tested some new markers, is not particularly new. The link between inflammation and AF has already been described. What would be really interesting and probably new is the testing for prediction of AF recurrences. Are you able to perform this kind of analysis. Please provide at least some potential use of your results for clinicians.
Author Response
Please, see the attachment.

Reviewer 2 Report
I have read with interest paper by Martins et al. about new inflammatory markers for patients with NVAF. Inflammation is definitely a part of AF pathophysiology, thus the research topic is relevant and authors identified novel inflammation-related molecules whose levels are increased in patients with AF. However, authors are presenting the study as a biomarker study – nevertheless, they do not present any single sensitivity or specificity values for their identified biomarkers and their inclussion/exclussion criteria makes generalization of the results challenging. Specific comments how to edit the manuscript can be found bellow.
Major concerns
1) Study is presented as a biomarker study, but showing that some molecule levels are increased in specific cohort of patients compared to those without the disease does not qualify the molecule to be a biomarker. Please, provide sensitivity/specificity values for the molecules that remained their relation to AF after multivariate analysis.
2) Authors are providing multivariate analysis in which they are adjusting the values of biomarkers for the basic comorbidities and laboratory values that were different between the control and diseased groups. Why in the first place the control group was not designed to contain the same amount of hypertensive and diabetes patients? Selecting proper and adequate control group for biomarker study is of utmost importance to show valid results. Control groups shall be age/sex/BMI/comorbidites matched.
3) Authors excluded those individuals using antiplatelet agents or non-steroidal anti-inflammatory drugs – this totally disables generalization of results as most of the patients with AF are already having coronary artery disease or they are taking antiplatelet drugs as the primary prevention. Also other exclussion criteria (like non involving those with chronic kidney disease) are disabling the generalization of results and the study shall be presented more like the pathophysiological study focusing on identification of novel inflammatorty molecules invovled Af pathophysiology than on novel biomarker discovery.
4) In the text and Table 2 authors are presenting levels of IL-2 as being statistically significant however p-value is 0,056 which is not bellow 0,05…
Minor concerns
1) In Table 2 results are presented as “2.54 . 1012“ or „9.58 . 1088“ which is very difficult to read, please, edit.
2) Results for IP-10 „1.01 (1.00-1.02) 0.046“ may be statistically but are hardly clinically significant as change in OR by 0,01 can hardly be of any predictive value.
Typos
There are several typos thorought the manusript that shall be carefully english proof-read.
Author Response
Please, see the attachment.

Round 2
Reviewer 1 Report
Thank you very much for having addressed my comments. This is a nice little study provinding some new elements but without any big news. Scientifically it's is acceptable.
Author Response
We thank the reviewer for this/hers comments.
Reviewer 2 Report
I have read the revised version of the manuscript and the response to my comments. I appreacite authors explanations to my comments and most of them were meant as the guides for their future studies as I realized that now "when the paper is under peer-review" it is difficult to enroll new patients matched to the NVAF group. As they state, limitations are mentioned in the limitations, multivariate modelling was performed, thus these comments I accept as answered.
However, in the abstract, author state "We aimed to evaluate a range of inflammatory parameters in individuals with nonvalvular AF (NVAF) as potential disease biomarkers" - I thereby ask authors again, to provide (even as a supplementary material) the sensitivity and specificity analysis for their newly identified biomarkers, OR delete the statement about the biomarkers from the abstract (and thorough the manuscript) and focus the manuscript solely on identification of novel inflammatory markers potentialy involved in the pathophysiology of NVAF, which I believe is more suitable, based on the study desing and identification of IP-10.
Further, after repeated reading of the manuscript, I would like to ask authors to comment on statistics as they only state that T-test and Mann-Whitney test were used based on data distribution. However, after detailed study of the graphs, it is obvious, that some individuals in some parameters were exploiting something called "far values" (even in Figure 1, part G, IP-10 levels vere very high in 1 patients in non-NVAF group and in several patients of NVAF group) - I would like to know whether for the statistical purposes, the "far values" were extrapolated or removed, as these far values may totally alter the results and mentioned test may not be suitable (they may not work), if far values are present.
Thank you for considering my further comments.
Author Response
We thank the reviewer´s first comment and agree the expression “as potential disease biomarker” must be deleted/changed from the manuscript. Actually, this should have been be done by the first round of this review process.
As for the second reviewer´s comment, outliers were not extrapolated or removed from analyses. Instead, we used a nonparametric (Mann-Whitney) test to assess differences between groups. Nonparametric tests are ranks based, i.e., the relative position of an individual compared to others, and they are not affected by extreme values. In other words, ranks and medians are more robust to outliers.
First, the Shapiro-Wilk normality test was used to test the normality of data. We used the Student’s t-test if the data followed a normal distribution and the Mann–Whitney U test if the data were determined not to follow a normal distribution (as described in page 4, Methods section, 2.5 Statistical Analysis). T-test was used to test the group differences regarding total cholesterol, LDL, and HDL levels (table 1). All the other variables (including biomarker levels) were determined not to follow a normal distribution. As duly observed by the reviewer, biomarker data were skewed. Therefore, the Mann–Whitney U test was used to test group differences in all biomarkers (please see figure captions), and outliers were not removed.